# Historical Eco-Environmental Quality Mapping in China with Multi-Source Data Fusion

Shaoteng Wu [1], Lei Cao [1,*], Dong Xu [2,3,4] and Caiyu Zhao [5]

1    School of Architecture, Tianjin University, Tianjin 300072, China
2    State Key Laboratory of Remote Sensing Science, Faculty of Geographical Science, Beijing Normal University, Beijing 100875, China; xd@mail.bnu.edu.cn
3    Institute of Remote Sensing Science and Engineering, Faculty of Geographical Science, Beijing Normal University, Beijing 100875, China
4    Key Laboratory of Environmental Change and Natural Disasters of Ministry of Education, Beijing Normal University, Beijing 100875, China
5    Electronic Information Engineering College, Changchun University, Changchun 130022, China; zhaocy0530@163.com
*    Correspondence: 13260821710@163.com

**Abstract:** Since the initiation of economic reforms and opening up, China has witnessed an unprecedented rate of development across all sectors. However, the country has also experienced severe ecological damage, surpassing that of many other nations. The rapid economic growth has come at the expense of the environment, revealing a significant lack of coordination between urbanization and eco-environmental protection in China. Consequently, there is an urgent need for a comprehensive and continuous historical dataset of China's eco-environmental quality (EEQ) based on remote sensing, allowing for the analysis of spatial and temporal changes. Such data would provide objective, scientific, and reliable support for China's eco-environmental protection and pollution prevention policies, while addressing potential ecological risks resulting from urbanization. To achieve this, the entropy value method is employed to integrate multi-source remote sensing data and construct an evaluation system for China's EEQ. Historical data from 2000 to 2017 is plotted to illustrate China's EEQ over time. The findings of this study are as follows: (1) The entropy method effectively facilitates the construction of China's eco-environmental quality assessment system. (2) From 2000 to 2017, approximately 39.7% of China's regions witnessed a decrease in EEQ, while 60.3% exhibited improvement, indicating an overall enhancement in EEQ over the past eighteen years. (3) The Yangtze and Yellow River basins experienced improved EEQ due to China's ecological restoration projects. (4) The future EEQ in China demonstrates a subtle positive trend across diverse contexts. This study departs from conventional approaches to EEQ evaluation by leveraging the advantages of multivariate remote sensing big data, including objectivity, timeliness, and accessibility. It provides a novel perspective for future eco-environmental quality evaluation.

**Keywords:** eco-environment quality; entropy method; remote sensing; China



## 1. Introduction

Since the beginning of the twenty-first century, global society has undergone progress and technological advancements, while China has made significant strides in urbanization and industrialization [1,2], However, this development has resulted in a progressive degradation of the eco-environmental quality (EEQ) [3,4]. It is essential to recognize that social progress and human development are intrinsically linked to the ecological environment [5]. In the 1990s, the United Nations organized conferences such as the "Conference on Environment and Development" and the "Habitat Conference" [6], marking the gradual emergence of ecological environment research as a significant area of focus for scholars. Additionally, with societal progress, residents' economic income and living standards have

improved, leading to an inevitable pursuit of a high quality of life, which necessitates a sound ecological environment as a fundamental prerequisite [7]. Given the growing global awareness of the ecological environment, the evaluation and protection of the ecological environment have become crucial tasks in China's modernization endeavors.

Since the reform and opening up, China has witnessed unprecedented development across all fields [8]. However, simultaneously, the ecological environment in China has suffered more severe damage compared to other countries worldwide [9]. The rapid economic development has come at the expense of ecological degradation, highlighting a disconnect between urbanization development and ecological protection in China, thereby hindering the achievement of sustainable development goals [10,11]. This low-coupling phenomenon is a growing concern for both the state and society, particularly as the extent of ecological damage increasingly affects people's lives. In 2000, China introduced the National Outline of Ecological Environmental Protection [12], which emphasized the crucial environmental policy of "equal emphasis on pollution prevention and ecological protection," thereby urging nationwide ecological.

When it comes to assessing EEQ, the OECD (Organization for Economic Cooperation and Development) introduced the "pressure-state-response" (PSR) model in 1990, which was the world's first recognized EEQ assessment index system [13]. As scientific progress continues, cross-disciplinary applications are becoming increasingly common. For instance, Mohamed et al. [14] utilized GIS and RS to examine the interplay between ecological factors, EEQ, and urbanization in the Nile Delta, revealing that rapid urbanization poses significant threats to both ecology and EEQ. Ducrot R et al. [15] employed 3S technology to analyze the reciprocal relationship between EEQ and urbanization. Sanna [16] employed principal component analysis (PCA) and cluster analysis to investigate the impact of urbanization development on EEQ in Helsinki. Ja-Hyum Kim et al. [17] devised a novel modeling approach for evaluating and monitoring urban ecological safety. In the twenty-first century, studies focusing on EEQ evaluation have become increasingly rigorous and comprehensive [18]. Some scholars have analyzed EEQ from social, economic, and natural perspectives [19]. For instance, Tan Zifang et al. [20] constructed an urban ecological suitability index model based on three dimensions (living, production, and environment) and applied it to analyze the EEQ of Changsha city, providing scientific and empirical support for environmental protection policies. Additionally, ecological niche theory has been employed by scholars [21] to assess EEQ. Bai Jie [22] conducted a quantitative analysis of the ecological niche in 14 cities in Gansu Province in 2009, employing dynamic cluster analysis. Existing studies on EEQ evaluation have primarily focused on the comprehensive evaluation process, including indicator selection, evaluation principles, and models [23]. However, there is a notable lack of an internationally accepted standard with high precision for EEQ evaluation [24].

In conclusion, there is a scarcity of studies on EEQ evaluation in China, and most of the existing studies are limited to specific regions and time periods. Furthermore, there is a dearth of research on continuous, long-term monitoring of EEQ in China since the twenty-first century, which is necessary to comprehensively analyze spatial and temporal changes at a nationwide scale from both horizontal and vertical perspectives. Additionally, many current studies on EEQ evaluation rely on statistical data, which can introduce subjectivity due to the uncertainties associated with different statistical sources. Furthermore, traditional EEQ evaluation methods based on statistical data lack the ability to offer insights into the spatiotemporal changes of future EEQ. Conversely, remote sensing data possesses distinct characteristics of objectivity, scientific rigor, and timeliness. Hence, there is a pressing need to develop a regional EEQ evaluation system that integrates multi-source spatial data, thereby addressing the limitations of current research approaches. This will require a continuous, long-term series of EEQ data for China to analyze trends and spatial variability objectively, scientifically, and reliably. Such data support is crucial for informing China's ecological environmental protection and pollution prevention policies and mitigating potential ecological risks resulting from urbanization in the future.

Here, based on the evaluation guidelines proposed by the National Environmental Monitoring Centre of China, namely, comprehensive principles, representative principles, scientific principles, comparability principles, and operability principles, we integrated multi-source spatial data to develop a comprehensive evaluation system for EEQ, known as the Modified Remote Sensing Eco-Environmental Quality Index (M-RSEQI). This evaluation system is designed to possess spatial and temporal universality, achieved through the application of the entropy method to ensure objectivity. By utilizing this system, we aim to monitor the continuous, long-term spatial and temporal variations of EEQ across China. The primary objective of this endeavor is to accurately and objectively depict the actual ecological environment of China based on scientific principles.

## 2. Materials and Methods

### 2.1. Study Area

China has a complex topography (Figure 1) and a wide variety of ecosystems [23], encompassing subtropical, tropical, temperate, boreal, and highland regions. Among these, forest and grassland ecosystems dominate in terms of extent. Forest ecosystems span across Northeast, North, South, Southwest, and Northwest China, covering approximately 22.96% of the country's total area. Meanwhile, grassland ecosystems thrive in the Qinghai-Tibet Plateau, Inner Mongolia, Xinjiang, and Northwest China, accounting for 41.94% of the nation's land area [24].

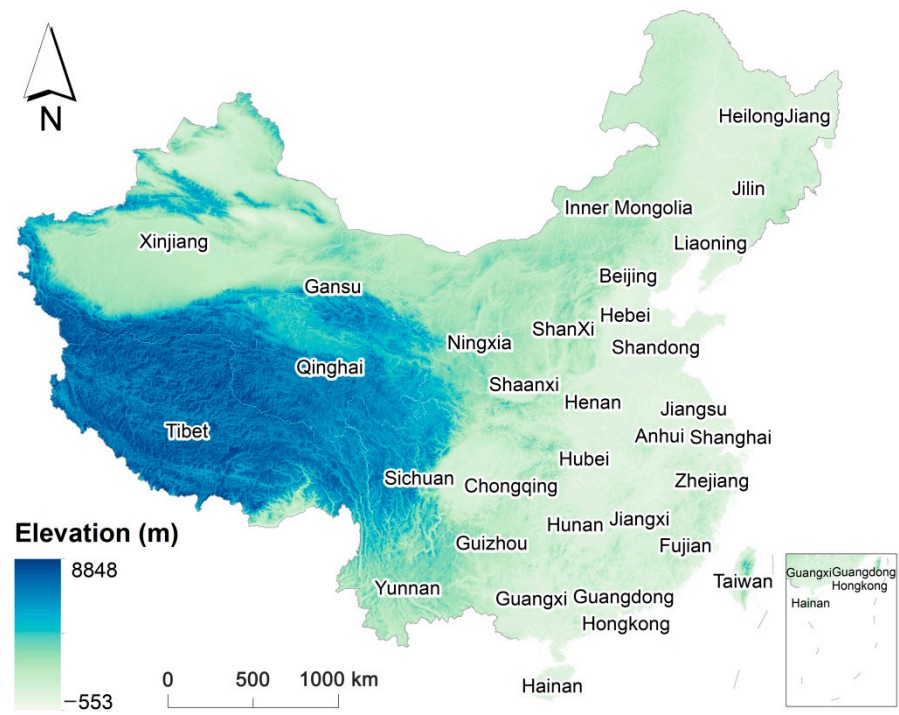

**Figure 1.** Topographical map of China and the distribution of the different provinces.

In addition to forests and grasslands, China boasts a variety of other ecosystems, including wetlands, deserts, islands, and marine habitats. Its wetlands occupy a vast expanse of 535,300 square kilometers, ranking among the world's largest [25]. The country's deserts span an area of approximately 100,000 square kilometers and encompass notable regions such as the Taklamakan Desert, Kubuqi Desert, and Ordos Desert. Furthermore, China is home to numerous islands such as Taiwan, Hainan, Dongsha Islands, Spratly Islands, and Diaoyu Islands. Marine ecosystems envelop the East China Sea, South China Sea, and Yellow Sea, with the East China Sea representing the largest marine ecosystem in China.

From a socio-economic perspective, China stands as the world's most populous country [26]. It also maintains its position as the second-largest economy globally, with a total national GDP of RMB 134.9 trillion in 2022 and an economic growth rate of 5.2%. China possesses a substantial consumer market and export potential, projected to reach RMB 80 trillion and RMB 20 trillion, respectively, by 2030 [27].

In summary, China harbors a vast range of ecosystems and presents significant socio-economic prospects. Moving forward, it is crucial for China to prioritize ecological and environmental preservation as a foundation for achieving sustainable economic growth.

### 2.2. Datasets

The data in this paper are all remote sensing data (Table 1), including precipitation (PRE), temperature (TEMP), net primary productivity (NPP), vegetation coverage (VC), digital elevation model (DEM), land-use and cover change (LUCC), population (POP) and potential evapotranspiration (PET).

**Table 1.** Detailed description of the data.

| Data Name | Time Range | Spatial Resolution | Time Resolution | Source |
|---|---|---|---|---|
| Precipitation | 2000–2017 | 1000 m | Monthly | NTPDC [a] |
| Temperature | 2000–2017 | 1000 m | Monthly | NTPDC [a] |
| Net primary productivity (MOD17A3) | 2000–2017 | 1000 m | Annual | NASA [b] |
| Vegetation coverage (MOD13A2) | 2000–2017 | 250 m | 16-day | NASA [b] |
| DEM (SRTM) | 2000–2017 | 250 m | — | USGS [c] |
| Land-use and cover change (MCD12Q1) | 2001–2017 | 500 m | Annual | NASA [b] |
| Population (Landscan) | 2000–2017 | 1000 m | Annual | ORNL [d] |
| Potential evapotranspiration (MOD16A3) | 2000–2017 | 500 m | 8-day | NASA [b] |
| Land-use and cover change (CLUD) | 2000 | 1000 m | Annual | CAS [e] |
| CMIP 6 | 2020–2100 | 1° | Annual | NASA [b] |
| Predicting LUCC | 2020–2100 | 1000 m | Annual | Paper [28] |
| Predicting POP | 2020–2100 | 1000 m | Annual | Paper [29] |

[a] National Tibetan Plateau Data Center (https://data.tpdc.ac.cn/en/news, accessed on 1 March 2023). [b] National Aeronautics and Space Administration (https://www.nasa.gov/, accessed on 1 March 2023). [c] United States Geological Survey (https://www.usgs.gov/, accessed on 1 March 2023). [d] Oak Ridge National Laboratory (https://www.ornl.gov/, accessed on 1 March 2023). [e] Chinese Academy of Sciences (http://www.resdc.cn/, accessed on 1 March 2023).

The data processing process mainly includes annual synthesis of indicator data, resampling, indicator standardization (selection of uniform extreme values), and weighted summation of indicators (Figure 2). The details are shown below:

(1) Synthesis of annual data: MOD17A3 (NPP) data [30] are annual data synthesized by averaging the monthly data. MOD13A2 (NDVI) data [31] are annual data synthesized by averaging the growing season data. MCD12Q1 (LUCC) data [32] are annual data synthesized by averaging the monthly data. MOD16A3 (PET) data [33] are annual data synthesized by averaging the monthly data. Precipitation (PRE) data [34] are annual data synthesized by summing the monthly data. Temperature (TEMP) data [34] are annual data synthesized by averaging the monthly data.

(2) Projection transformation: MRT software was used to perform the projection transformation NPP, NDVI, LUCC and PET data and unify them into the WGS 1984 geographic coordinate system.

(3) Resampling: all data was resampled to 1000 m resolution.

(4) Data format transformation: MRT software was used to perform data format transformation of NPP, NDVI, LUCC and PET data, and HDF was transformed to TIFF

format. In addition, Matlab was used to transform the data format of PRE and TEMP from NETCDF (.nc) to TIFF.

(5) Data stitching and cropping: 19 sheets MODIS data covering the Chinese region were stitched together.

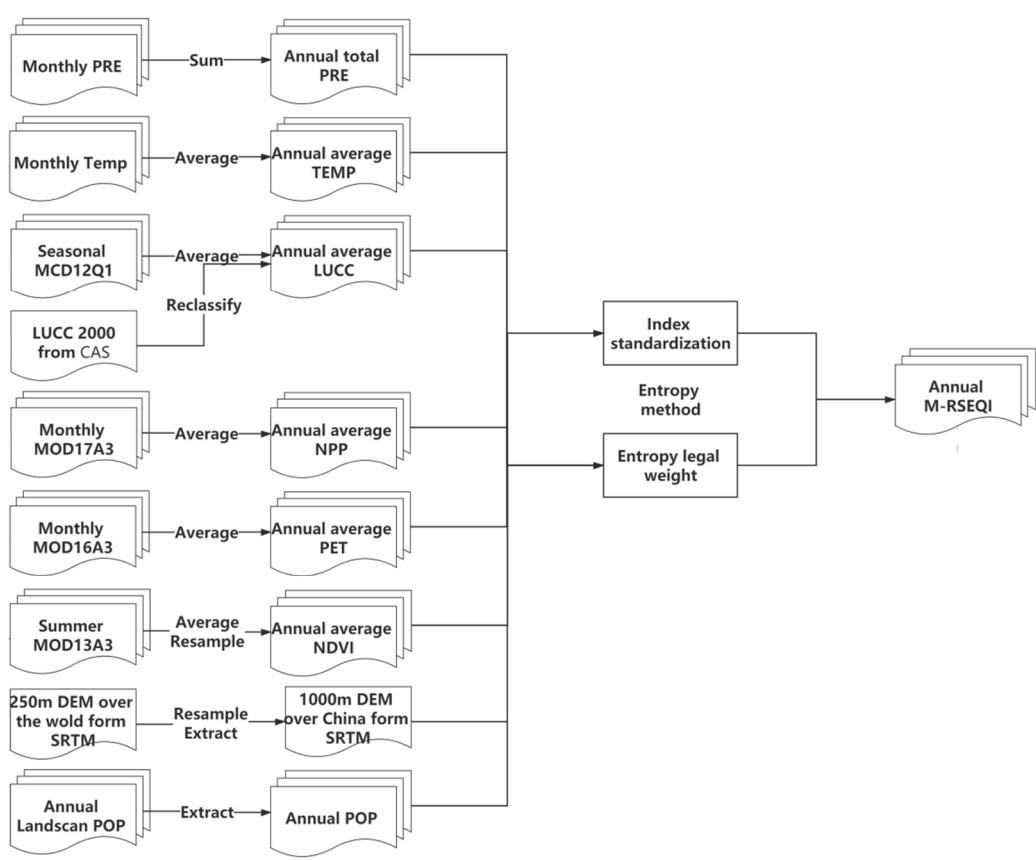

**Figure 2.** M-RSEQI calculation flow chart.

*2.3. Assessment of China's EEQ*

2.3.1. Principle of Indicator Selection

Regarding the evaluation of EEQ, several principles have been put forward by scholars for the selection of evaluation indicators. Xu Yan emphasized that evaluation indicators should be scientific, dominant, practical, and comprehensive. Wang suggested that the evaluation of EEQ should be ecologically sound and meet scientific standards. Zhang Hong proposed following principles such as importance, typology and regionality, sustainable use, and recognizing the value of noble ecosystems and priceless natural resources. Furthermore, the China General Environmental Monitoring Station highlighted the importance of indicators being representative, comprehensive, concise, comprehensive, and easily applicable. These principles provide valuable guidance for the development and selection of evaluation indexes in the assessment of EEQ [35].

In this study, we adopted the principles of indicator selection proposed by the China General Environmental Monitoring Station and carefully selected eight indicators that take into account the specific ecological environment of China and factors such as data accessibility.

To further assess the appropriateness of the selected indicators, we conducted a covariance diagnostic analysis using the linear regression analysis method to determine if there was any redundancy among the eight indicators [36]. The variance inflation factor (VIF) and tolerance (TOL) are commonly used indicators for diagnosing multicollinearity [37]. VIF and TOL are reciprocals, and if TOL > 0.1 or VIF < 10, it indicates that the selected evaluation indices are reasonable.

Since the indicators for each year are independent, we conducted the diagnostic analysis using the 2017 data in this study. The results, presented in Table 2, demonstrate that there was no information redundancy among the eight selected indicators. This indicates that the indicators chosen for this study were appropriate and do not exhibit multicollinearity issues.

**Table 2.** Index collinearity diagnosis results.

| Index | 2017 | |
| | Variance Inflation Factor | Tolerance |
| | VIF | TOL |
|---|---|---|
| PRE | 8.097 | 0.123 |
| TEMP | 9.411 | 0.106 |
| NPP | 5.758 | 0.174 |
| NDVI | 5.922 | 0.169 |
| DEM | 2.975 | 0.336 |
| LUCC | 4.095 | 0.244 |
| POP | 1.349 | 0.741 |
| PET | 2.634 | 0.380 |

2.3.2. Calculation of EEQ

Different indicators possess distinct dimensions, and thus, it is necessary to convert each indicator into a dimensionless form prior to evaluating the EEQ. In this study, the extreme difference standardization method was employed to transform the eight indicators. These indicators were categorized into positive and negative based on their respective impacts on the EEQ. Positive indicators encompassed PRE (precipitation), NPP (net primary productivity), NDVI (normalized difference vegetation index), DEM (digital elevation model), and LUCC (land use and land cover change), all of which contribute to the enhancement of the ecological environment. Conversely, negative indicators, including TEMP (temperature), POP (population), and PET (potential evapotranspiration), denote factors that are ecologically detrimental. The standardized formulas employed are as follows.

Positive Indicators:

$$r_s^+ = (I_j - I_{\min})/(I_{\max} - I_{\min}) \tag{1}$$

Negative indicators:

$$r_s^- = (I_{\max} - I_j)/(I_{\max} - I_{\min}) \tag{2}$$

where, $r_s^+$ is the standardized value of the $j$th indicator; $I_j$ is the initial value of the $j$th indicator; $I_{\min}$, $I_{\max}$ are the minimum and maximum values of the $j$th index.

The indicator weighting model serves as a crucial aspect of the evaluation system, as the chosen model's merits and limitations can significantly influence the evaluation outcomes. Currently, subjective and objective methods are commonly employed for determining the weights. The subjective approach encompasses methods such as the gray relation method (GRM) [38] and the analytic hierarchy process (AHP) [39]. On the other hand, objective methods include the entropy method (EM) [40] and the principal component analysis (PCA) [41]. The subjective method exhibits notable limitations in its application and lacks robust theoretical support, whereas the objective method calculates weights based on the interrelationships among indicators. To mitigate the subjective influence stemming from human factors, this study adopts the entropy value method to ascertain the weights of the eight indicators:

$$w_{ij} = \frac{r_{ij}^+}{\sum_{i=1}^n r_{ij}^+} \text{ or } w_{ij} = \frac{r_{ij}^-}{\sum_{i=1}^n r_{ij}^-} \tag{3}$$

$$e_j = -k \sum_{i=1}^{n} w_{ij} \ln w_{ij}, \; k = (\ln n)^{-1} \tag{4}$$

$$f_j = 1 - e_j \tag{5}$$

$$w_j = \frac{f_j}{\sum_{j=1}^{m} f_j} \tag{6}$$

where, $w_{ij}$ is the corresponding weight of the $i$th indicator of the $j$th city; $e_j$ is the entropy value of the $j$th indicator; $f_j$ is the redundancy factor of the $j$th indicator; $w_j$ is the weight corresponding to the $j$th indicator; In this paper, $i$ and $j$ are 369 and 8, respectively.

### 2.4. Trend Analysis

To examine the temporal and spatial changes in China's EEQ from 2000 to 2017, this study employs a one-dimensional linear regression model to calculate the trend of interannual variation in the M-RSEQI [41]. The slope derived from the model represents the interannual trend of the M-RSEQI. The calculation formula is presented below:

$$slope = \frac{n \times \sum_{i=1}^{n} (i \times MRSEQI_i) - \sum_{i=1}^{n} i \sum_{i=1}^{n} MRSEQI_i}{n \times \sum_{i=1}^{n} i^2 - \sum_{i=1}^{n} i} \tag{7}$$

where, *slope* is the slope in the one-dimensional regression equation of the linear fit of M-RSEQI with year for China 2000–2017; $i$ is the time variable and $n$ is the year; $n = 18$; $MRSEQI_i$ represents the multivariate remotely sensed EEQ in year $i$. When *slope* < 0, it indicates that the EEQ is decreasing. On the contrary, when *slope* > 0, it indicates that the EEQ is increasing. The absolute value of *slope* represents the rate of change of M-RSEQI, with a larger *slope* representing faster change and the opposite being slower.

## 3. Results

### 3.1. Spatial Pattern of EEQ in China

In order to gain insight into the degree of superiority and inferiority of China's EEQ from 2000–2017, this paper classified China's M-RSEQI from 2000–2017 based on Table 3, and the results are shown in Figures 3 and 4.

**Table 3.** M-RSEQI classification table based on natural breakpoint method.

| Criteria | ≥0.60 | 045~0.60 | 0.30~0.45 | 0.15~0.30 | ≤0.15 |
|----------|-------|----------|-----------|-----------|-------|
| Level | Excellent | Good | Average | Poor | Bad |

Based on the analysis of Figures 2 and 3, several conclusions can be drawn regarding China's EEQ from 2000 to 2017. Firstly, it is evident that a significant portion of China's EEQ fell into the "bad" grade, remaining stable at approximately 9 million km$^2$. Secondly, the regions categorized as "poor" grade spanned between 3 million km$^2$ and 4 million km$^2$. The subsequent grade is "average," while the areas designated as "good" or "excellent" were relatively small. Among the five grades, the "average" and "good" grades exhibited notable changes, while the remaining three grades showed insignificant variations. Regions with poor EEQ in China during the aforementioned period were primarily concentrated in the Beijing-Tianjin-Hebei region, Tacheng region, Turpan city, Bortala Mongol Autonomous Prefecture, Karamay city, Altai region, and Changji Hui Autonomous Prefecture. Conversely, areas with average or better EEQ were predominantly located in the Qinghai-Tibet Plateau, certain parts of Yunnan Province, and specific regions of Taiwan Province. Notably, the EEQ of the Beijing-Tianjin-Hebei region displayed significant improvement from 2000

to 2017, with a slight fluctuation in the area characterized by poor EEQ throughout this period. Overall, there was a considerable reduction in poor EEQ, closely associated with the industrial restructuring efforts undertaken in the Beijing-Tianjin-Hebei region [42,43].

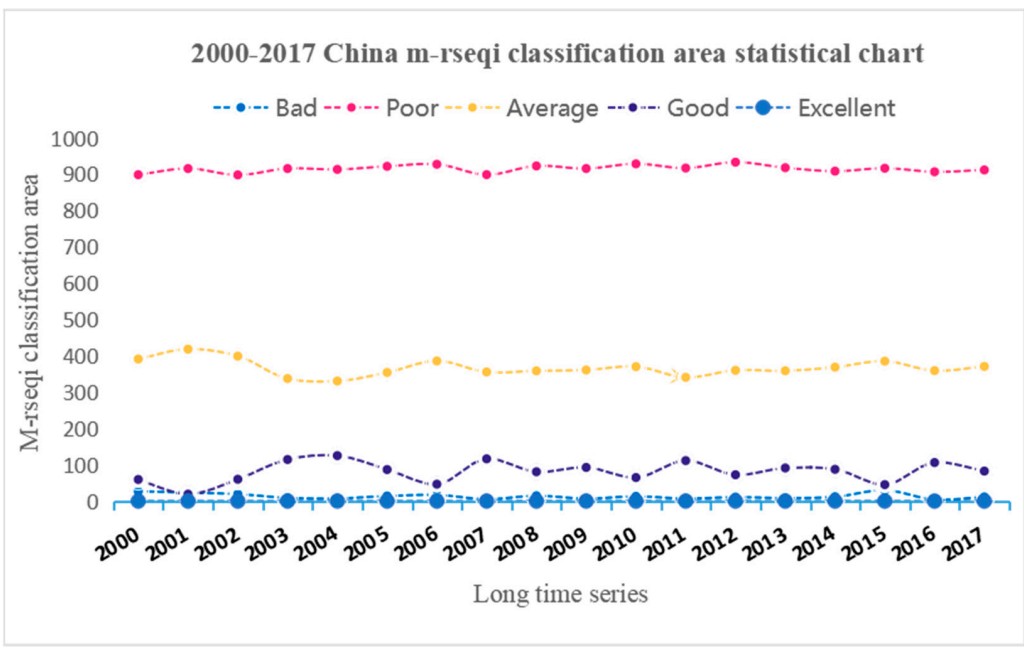

**Figure 3.** Area ($km^2$) of different EEQ categories in China from 2000 to 2017.

### 3.2. Spatial and Temporal Changes in China's EEQ

The spatial variations of the M-RSEQI trends in China from 2000 to 2017 are depicted in Figure 5 and Table 4. It is evident that the trends exhibit considerable diversity across different regions. The percentage of area where the M-RSEQI decreased and improved were 39.7% and 60.3%, respectively, indicating an overall improvement in EEQ across most regions of China over the past eighteen years. Notably, regions experiencing a faster improvement in the M-RSEQI accounted for 9.19% of China's total area (slope $\geq 2 \times 10^{-3}a^{-1}$). These regions primarily include the Loess Plateau, Qinghai Province, Ningxia Hui Autonomous Region, and areas surrounding Sichuan Province. Conversely, the regions where the M-RSEQI decreased encompassed the Tarim Basin in Xinjiang, northwest Inner Mongolia, parts of southern Tibet, Yunnan Province, and the coastal areas of the Yangtze River Delta and Pearl River Delta. These regions exhibited a slow declining trend.

Among the 35 provinces in China, including municipalities directly under the Central Government and Taiwan Province, 17 provinces displayed an increasing trend in the M-RSEQI, while 18 provinces exhibited a decreasing trend. Notably, the Ningxia Hui Autonomous Region had the fastest increasing speed (slope = $0.99 \times 10^{-3}a^{-1}$), whereas the Hong Kong Special Administrative Region had the fastest decreasing speed (slope = $-0.74 \times 10^{-3}a^{-1}$). At the national level, the M-RSEQI showed a gradual improvement trend, albeit at a slow pace (slope = $0.18 \times 10^{-3}a^{-1}$).

In Figure 6, it can be seen that China's EEQ did not change much during 2000–2017, and the overall EEQ was low, with the average value of M-RSEQI below 0.3 in all years. The best EEQ in China during 2000–2017 was in 2007, when several policy documents were introduced to prepare for the Olympics [44]. For example, regulations such as the "Opinions of the State Environmental Protection Administration on Further Strengthening Ecological Protection" and "Opinions on Strengthening Environmental Protection in Rural Areas" were introduced. Local governments organized environmental protection efforts to strengthen the fight against ecological and environmental violations. Heilongjiang Province also increased the intensity of law enforcement inspections of straw burning, greatly reducing traffic congestion, air pollution and forest fires caused by straw burning

and other accidents [45]. In addition, the worst EEQ in China was in 2001 and 2015. In 2001, during the golden period of China's economic and social development, the provinces experienced rapid economic growth. At the same time, many EEQ problems arose. Severe acid rain pollution existed in southeastern China. Acid rain occurs in more than 90% of the 188 cities that fall within the acid rain control area [46]. In the report released by the National People's Congress on the completion of environmental protection in 2015, it was pointed out that the problems of heavy environmental pollution, high environmental risks and ecological damage were particularly prominent in the country this year, thus showing that the ecological and environmental situation in China was particularly severe in 2015.

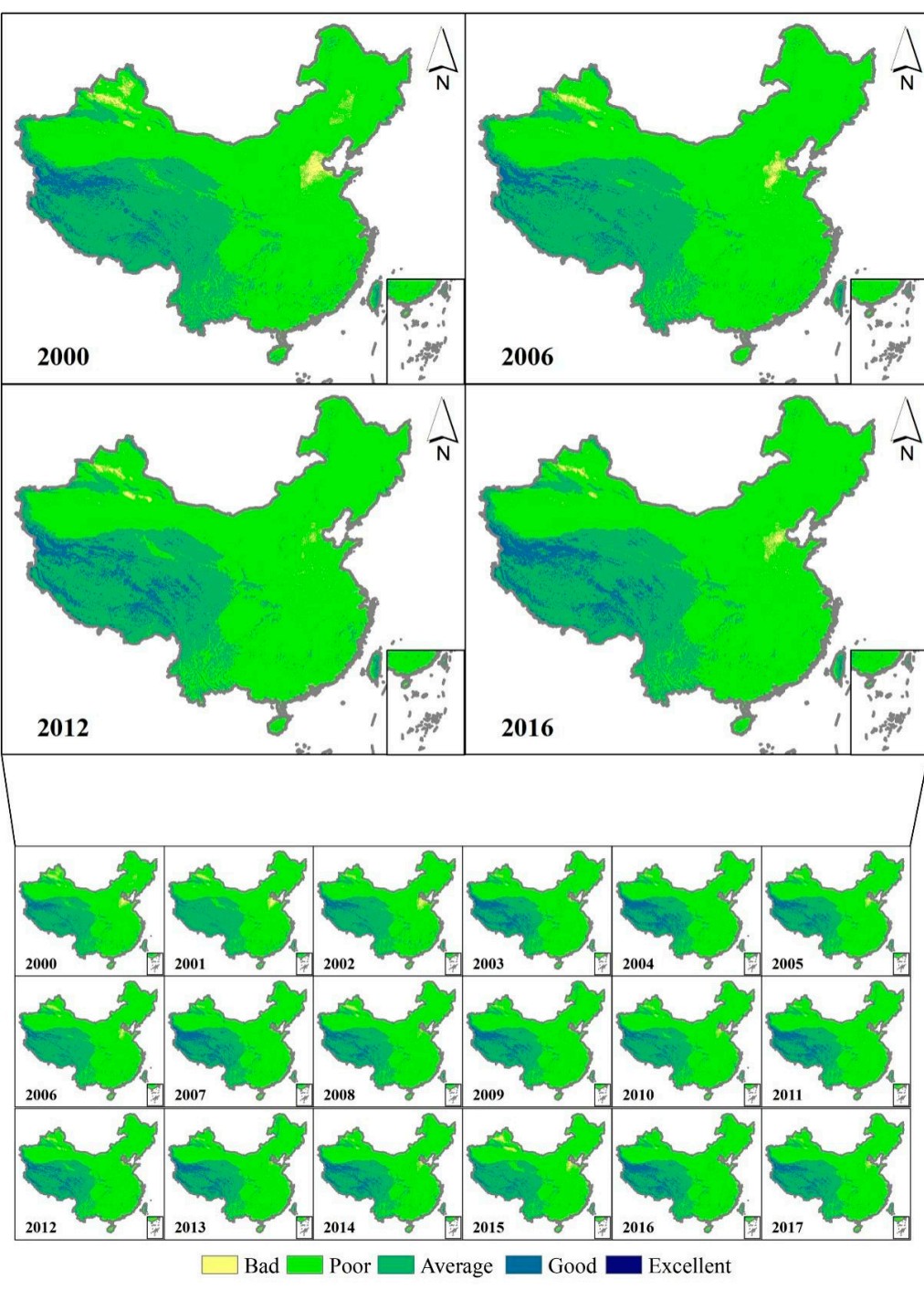

**Figure 4.** Spatial distribution pattern of M-RSEQI in China from 2000 to 2017.

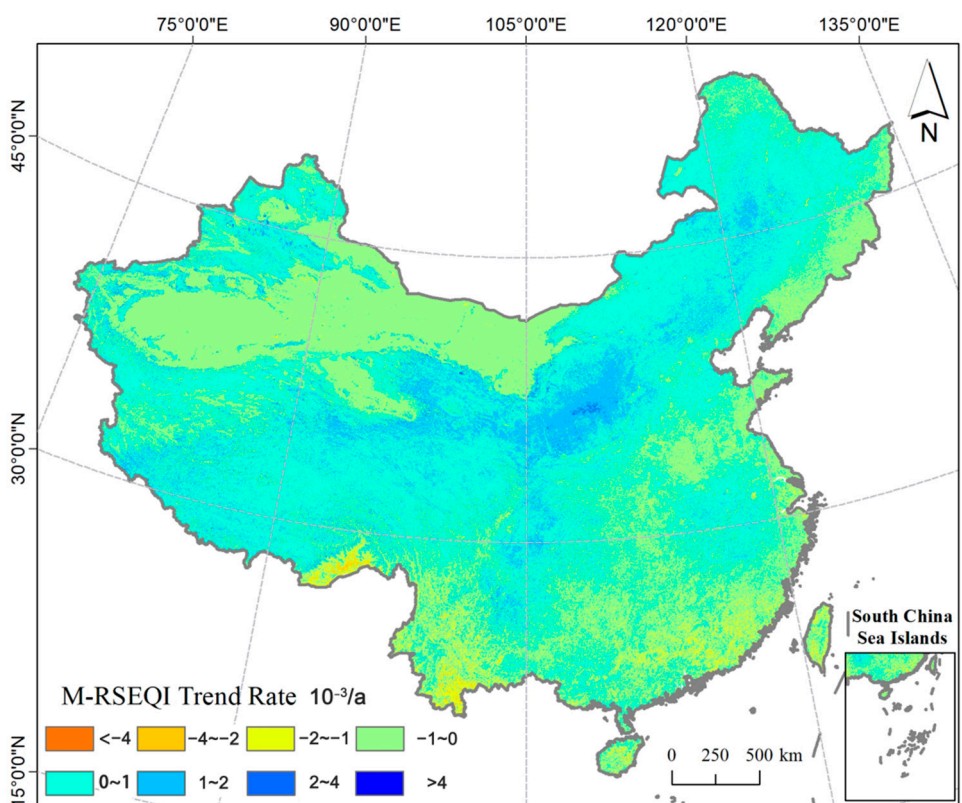

**Figure 5.** Change trend distribution of M-RSEQI in China from 2000 to 2017.

**Table 4.** M-RSEQI change trend classification standard and proportion of each level area.

| M-RSEQI | | | | | | | | |
|---|---|---|---|---|---|---|---|---|
| | <4 | −4~−2 | −2~−1 | −1~0 | 0~1 | 1~2 | 2~4 | >4 |
| Percentage | 0.02 | 0.24 | 2.50 | 36.94 | 51.11 | 8.77 | 0.42 | 0.00 |

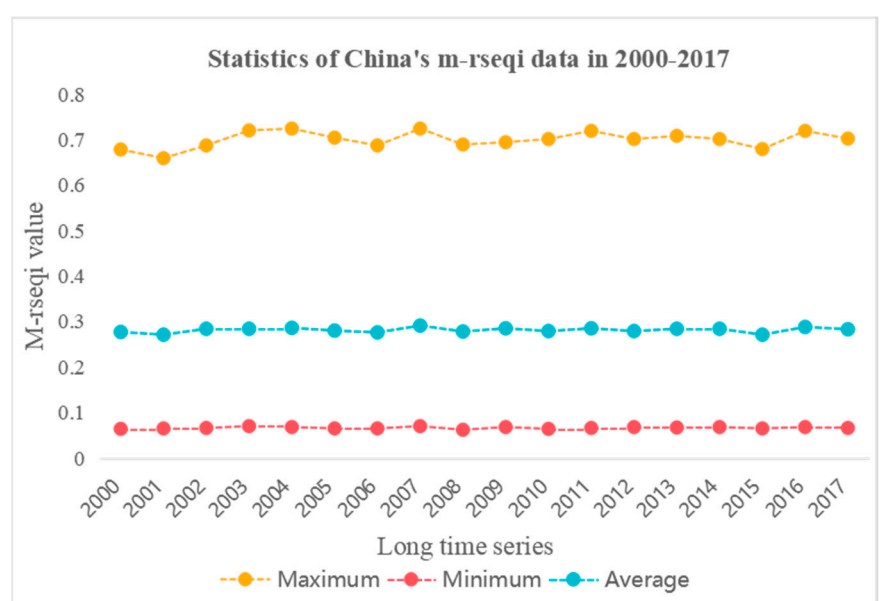

**Figure 6.** The time series analysis captures the changes in the maximum, minimum, and average values of M-RSEQI in China from 2000 to 2017.

Figure 6 illustrates the stability of China's EEQ during the period 2000–2017, with a consistently low overall EEQ, indicated by the M-RSEQI average value remaining below 0.3 throughout the years. The highest EEQ in China was observed in 2007, coinciding with the implementation of various policy measures in preparation for the Olympics [44]. Notably, regulations such as the "Opinions of the State Environmental Protection Administration on Further Strengthening Ecological Protection" and "Opinions on Strengthening Environmental Protection in Rural Areas" were introduced during this time. Local governments actively organized environmental protection efforts and reinforced measures to combat ecological and environmental violations. Additionally, in Heilongjiang Province, the enforcement of straw burning regulations was intensified, resulting in a significant reduction in traffic congestion, air pollution, and forest fires caused by straw burning and other incidents [45]. Conversely, the years 2001 and 2015 exhibited the lowest EEQ in China. In 2001, despite rapid economic growth during the country's golden period of economic and social development, several EEQ issues emerged. Southeastern China experienced severe acid rain pollution, affecting over 90% of the 188 cities falling within the acid rain control area [46]. Furthermore, the 2015 report released by the National People's Congress highlighted the prominent challenges of heavy environmental pollution, high environmental risks, and ecological damage, reflecting the severity of the ecological and environmental situation in China during that year.

### 3.3. EEQ of the Yangtze and Yellow River Basins

3.3.1. Monitoring of the EEQ of the Yangtze River Basin

The Yangtze River [47], known as the "mother river" of China, is the third longest river in the world and the first hydroelectric river. Spanning a length of 6,397 km, it traverses Qinghai, the Tibet Autonomous Region, Sichuan, Yunnan, Chongqing, Hubei, Hunan, Jiangxi, Anhui, Jiangsu, and Shanghai, playing a crucial role in China's energy, food, and economy. The Yangtze River basin is renowned for its abundant biological resources and stands as the most ecologically diverse region in China [48]. However, since the 21st century, the rapid economic development of the Yangtze River economic zone has led to a decline in the EEQ of the Yangtze River basin, necessitating long-term monitoring efforts [49].

As depicted in Figure 7, the distribution of EEQ in the Yangtze River basin in 2000, 2008, and 2017 exhibits a consistent pattern characterized by "worse in the east and better in the west." Certain regions in Chongqing consistently displayed poor EEQ over the three-year period, indicating insufficient implementation of ecological environmental protection measures in the area. Additionally, the Yangtze River Delta region also exhibited poor EEQ, highlighting the lack of synchronized development between urbanization and EEQ in the region, thereby demonstrating a low coupling phenomenon.

Figures 7 and 8 reveal that significant changes in EEQ primarily occurred in the Sichuan basin and certain areas of Chongqing city during the period of 2000–2017. Notably, Chongqing exhibited a larger region transitioning from poor to bad EEQ, suggesting efforts were made to improve the environmental quality. Conversely, the regions experiencing EEQ deterioration were concentrated in Sichuan Province, Hubei Province, and Shanghai.

According to Table 5, the majority of the Yangtze River basin, encompassing an area of 2,021,800 km$^2$, or 83.40% of the total, maintained an unchanged EEQ from 2000 to 2017. The area with decreased EEQ accounted for 144,400 km$^2$, equivalent to 5.96% of the total basin area. Conversely, the area with increased EEQ covered 257,900 km$^2$, representing 10.64% of the total basin area. Overall, the EEQ of the Yangtze River basin exhibited improvement.

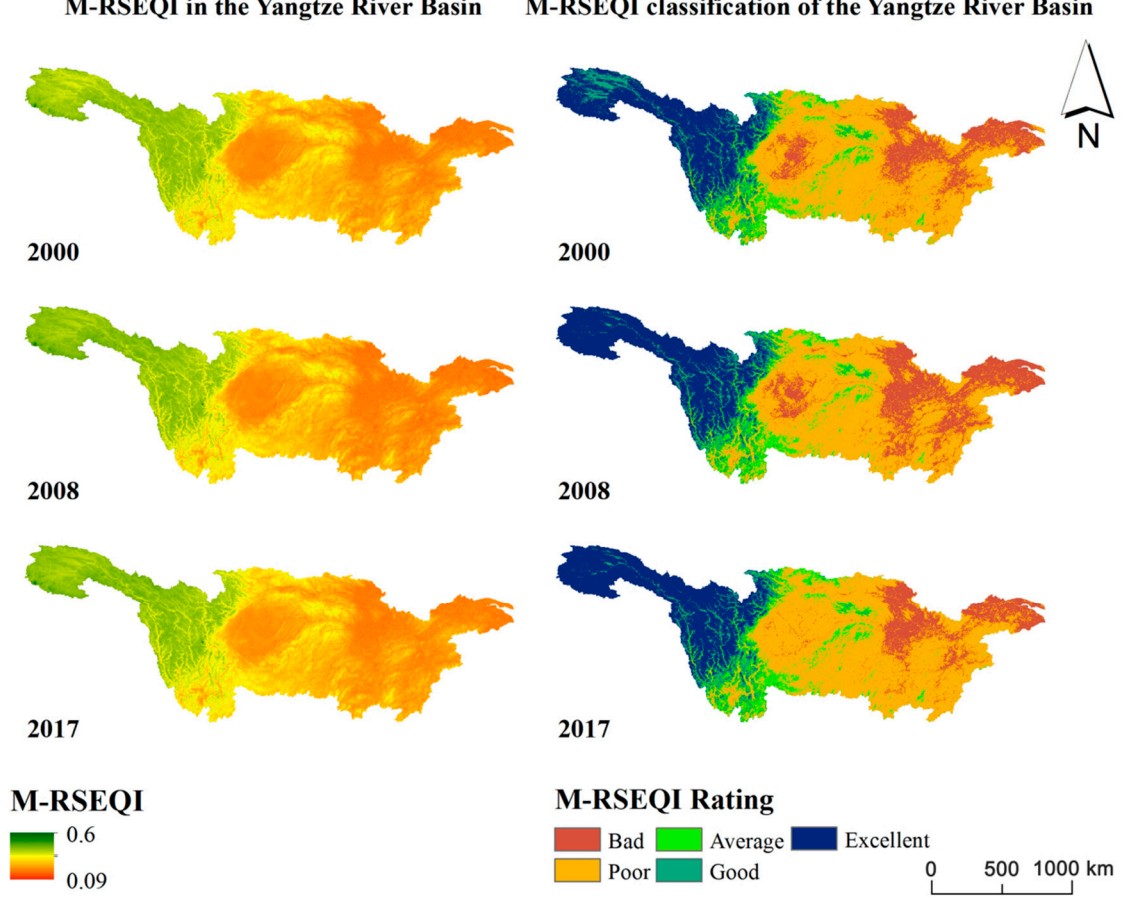

**Figure 7.** M-RSEQI distribution map and M-RSEQI classification map of the Yangtze River Basin in 2000, 2008 and 2017.

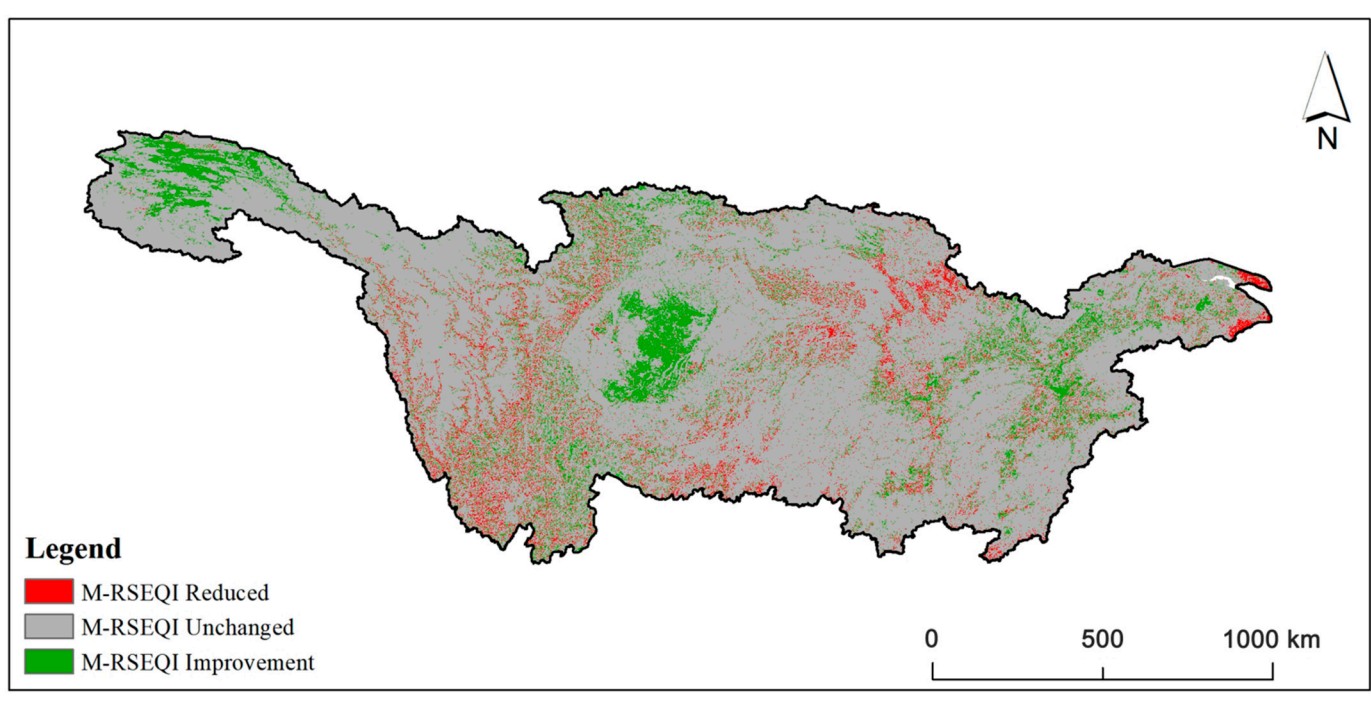

**Figure 8.** Change map of M-RSEQI in the Yangtze River Basin from 2000 to 2017.

**Table 5.** Change area of M-RSEQI in the Yangtze River Basin from 2000 to 2017.

| Level | Area (10,000 × km²) | Percentage (%) |
|---|---|---|
| M-RSEQI Reduced | 14.44 | 5.96 |
| M-RSEQI Unchanged | 202.18 | 83.40 |
| M-RSEQI Improvement | 25.79 | 10.64 |

### 3.3.2. Monitoring of the EEQ of the Yellow River Basin

The Yellow River basin, similar to the Yangtze River basin, serves as a vital economic belt and ecological safeguard for China [50]. However, extensive resource exploitation and agricultural reclamation have led to various ecological and environmental challenges in the Yellow River basin [51], such as the collapse of the overflow well in Luanchuan and excessive benzene content in Lanzhou's drinking water. Consequently, there is an urgent need to monitor the EEQ of the Yellow River basin and comprehensively study and analyze the ecological environment changes using scientific methods, aiming to establish a green economic zone along the Yellow River.

As illustrated in Figure 7, the distribution of EEQ in the Yellow River Basin remained similar in 2000, 2008, and 2017, exhibiting the characteristic pattern of "worse in the east and better in the west". Among the three years, Qinghai Province consistently displayed the highest EEQ, followed by Ningxia Hui Autonomous Region and Gansu Province. Conversely, Shaanxi Province, Shanxi Province, and certain regions of Inner Mongolia Autonomous Region consistently demonstrated lower EEQ throughout the same period.

Figures 9 and 10 illustrate that the changes in EEQ during the period of 2000–2017 were predominantly concentrated in the Inner Mongolia Autonomous Region, Shaanxi Province, and certain regions of Shanxi Province. These areas were characterized by a transition from a bad to poor EEQ rating. Additionally, the reduction in EEQ was minimal, suggesting a consistent improvement in the EEQ of the Yellow River basin over this period.

**Figure 9.** M-RSEQI distribution map and M-RSEQI classification map of the Yellow River Basin in 2000, 2008 and 2017.

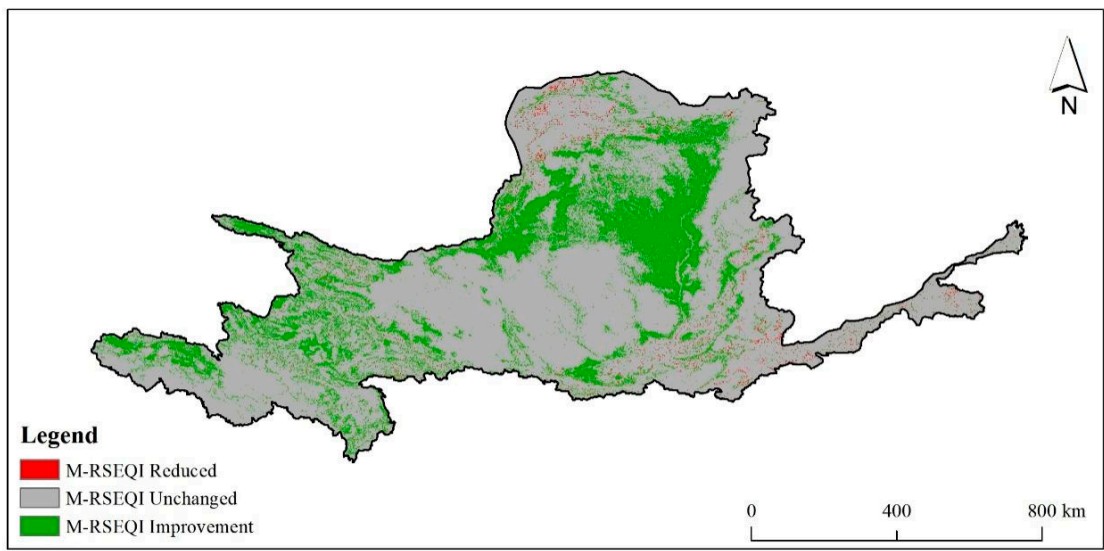

**Figure 10.** Change map of M-RSEQI in the Yellow River Basin from 2000 to 2017.

In Table 6, the majority of the Yellow River basin, encompassing an area of 879,400 km², accounting for 74.80% of the total basin area, exhibited an unchanged EEQ from 2000 to 2017. The area where the EEQ had decreased was 10,000 km², representing 0.86% of the total basin area. Conversely, the area where the EEQ had increased spanned 286,100 km², accounting for 24.34% of the total basin area. Overall, the EEQ of the Yellow River basin showed a consistent trend of improvement.

**Table 6.** Change area of M-RSEQI in the Yellow River Basin from 2000 to 2017.

| Level | Area (10,000 × km²) | Percentage (%) |
| --- | --- | --- |
| M-RSEQI Reduced | 1.00 | 0.86 |
| M-RSEQI Unchanged | 87.94 | 74.80 |
| M-RSEQI Improvement | 28.61 | 24.34 |

*3.4. EEQ of Major Cities*

Beijing, Shanghai, Guangzhou and Nanjing are important cities in China. Beijing is the capital of China and the political, cultural and diplomatic center of the country. Shanghai is the economic center of China and the frontier of reform and opening up [52]. Nanjing is the capital city of Jiangsu Province, known as the ancient capital of the Six Dynasties and an important national transportation hub [53]. Guangzhou is the capital city of Guangdong Province, a national mega-city, an important port for foreign exchanges, and a center of gravity for international trade [54]. The rapid economic and social development of the four major cities has also driven the development of the surrounding cities, and the Beijing-Tianjin-Hebei, Yangtze River Delta and Pearl River Delta regions are all using these four cities to maintain linkage development. Since the twenty-first century, ecological and environmental problems in the Beijing-Tianjin-Hebei, Yangtze River Delta and Pearl River Delta regions have been a hot topic [55], and it has become particularly important to use scientific and objective methods to study the EEQ of these regions in a long time series, which is very important for the governments in the region to formulate ecological protection.

Based on the 2000, 2008 and 2017 M-RSEQI indices, 30m impervious surface data from Tsinghua University and Google historical images, the coupling pattern between EEQ and impervious surface was investigated in Beijing, Shanghai, Guangzhou, and Nanjing. In Figures 11–14, it can be seen that Guangzhou had the best overall EEQ, followed by Beijing, and Nanjing and Shanghai have poorer EEQ. Specifically, the EEQ of Shanghai was the worst.

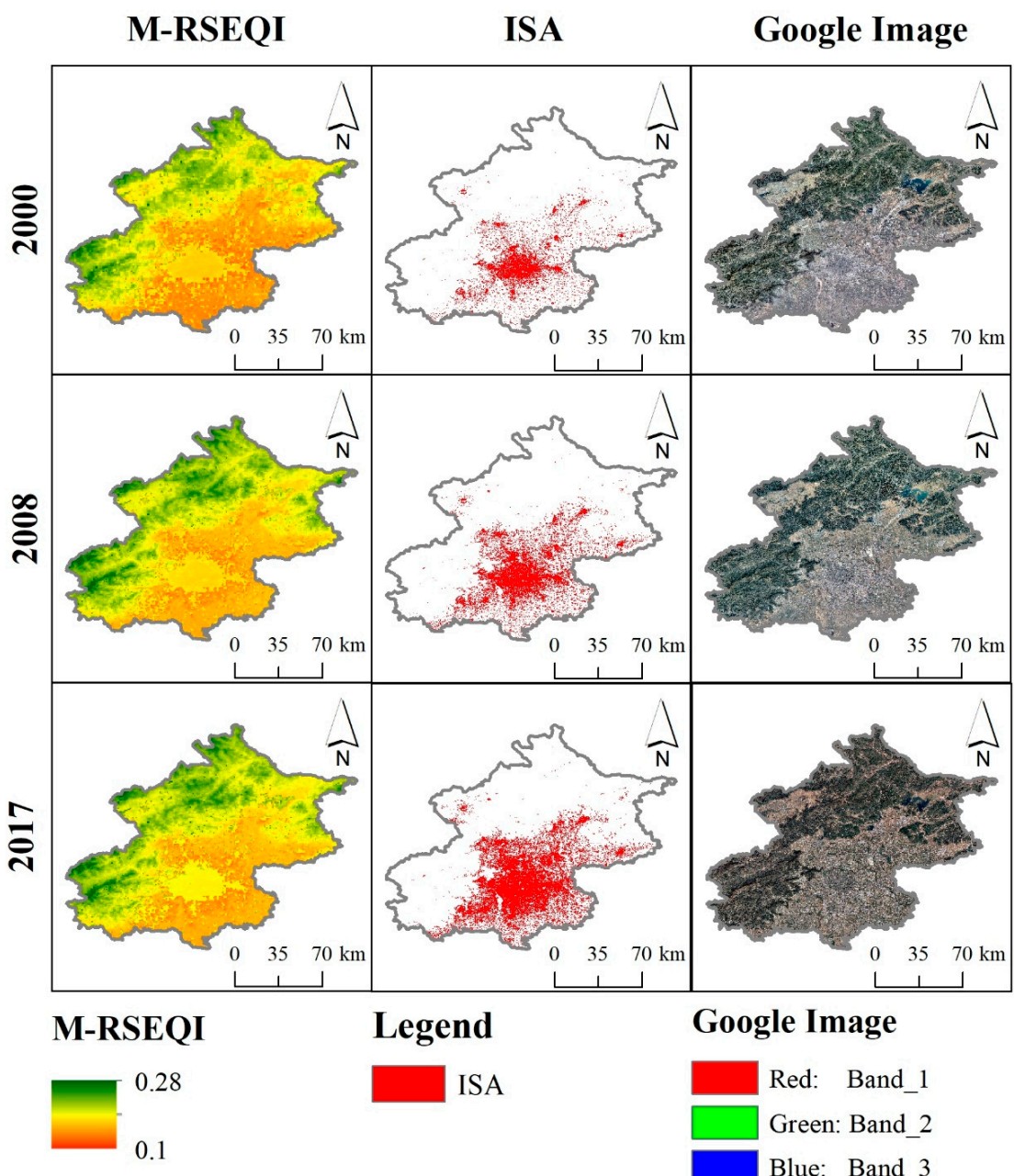

**Figure 11.** M-RSEQI, impervious area (ISA) and Google historical images of Beijing in 2000, 2008 and 2017.

Beijing, Shanghai, Guangzhou, and Nanjing are significant cities in China. Beijing serves as the capital and acts as the political, cultural, and diplomatic center of the country. Shanghai holds the role of China's economic hub and stands at the forefront of reform and opening up policies [51]. Nanjing, known as the ancient capital of the Six Dynasties, serves as the capital city of Jiangsu Province and plays a crucial role as a national transportation hub [52]. Similarly, Guangzhou, the capital city of Guangdong Province, holds the status of a national mega-city, serves as a prominent port for foreign exchanges, and acts as a center for international trade [53]. The rapid economic and social development of these four major cities has had a significant impact on the surrounding areas. The Beijing-Tianjin-Hebei, Yangtze River Delta, and Pearl River Delta regions all utilize these cities to drive and maintain interconnected development. Ecological and environmental issues in these regions have become a prominent topic since the 21st century [54]. Therefore, it is crucial

to employ scientific and objective methods to study the long-term EEQ trends in these regions. This research is of great importance for regional governments in formulating effective ecological protection strategies.

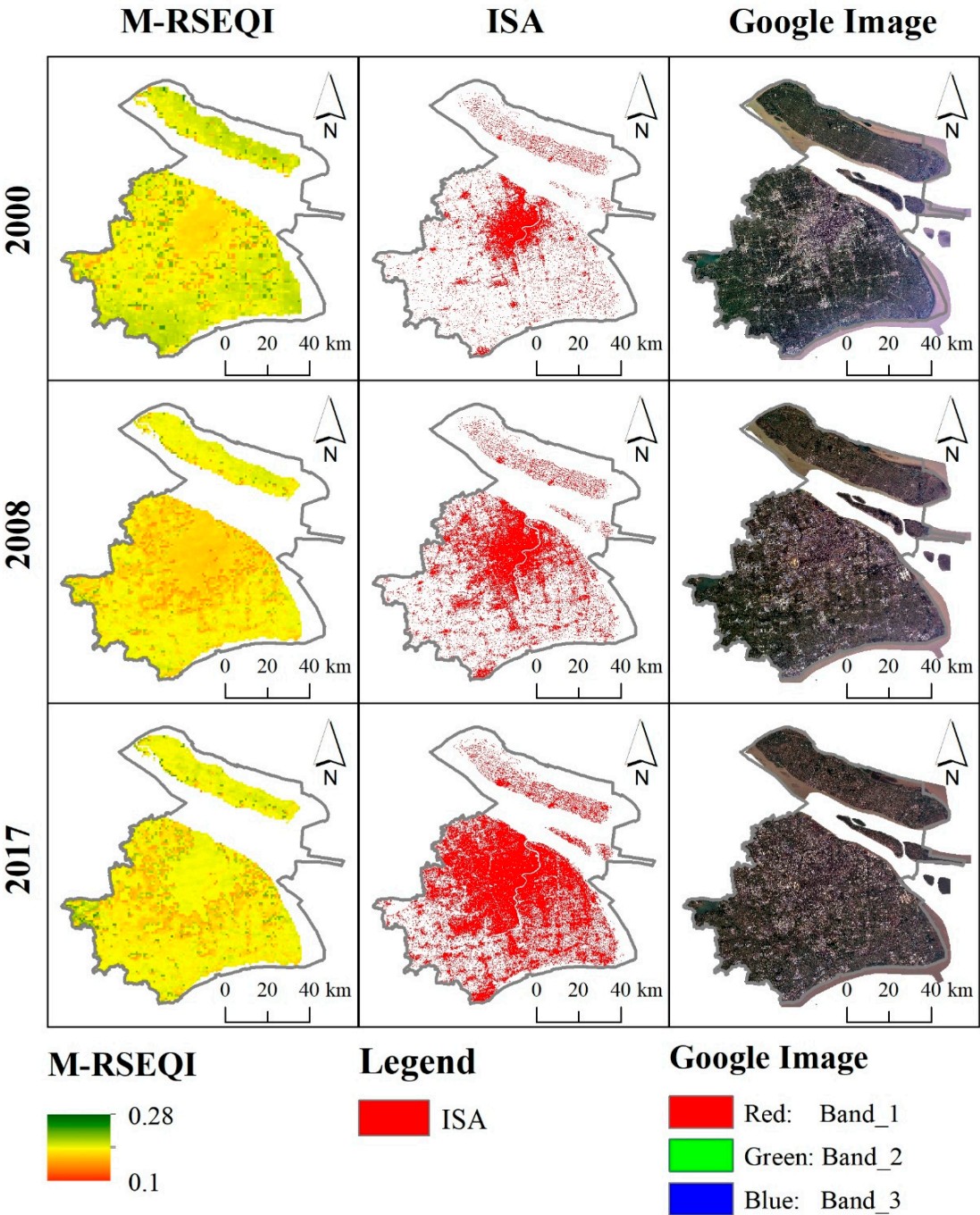

**Figure 12.** M-RSEQI, ISA and Google historical images of Shanghai in 2000, 2008 and 2017.

Using the 2000, 2008, and 2017 M-RSEQI indices, coupled with 30 m impervious surface data from Tsinghua University and historical images from Google, the relationship between EEQ and impervious surface was examined in Beijing, Shanghai, Guangzhou, and Nanjing. Figures 11–14 illustrate that Guangzhou exhibited the highest overall EEQ, followed by Beijing, while Nanjing and Shanghai had relatively poorer EEQ. Specifically, Shanghai had the lowest EEQ rating.

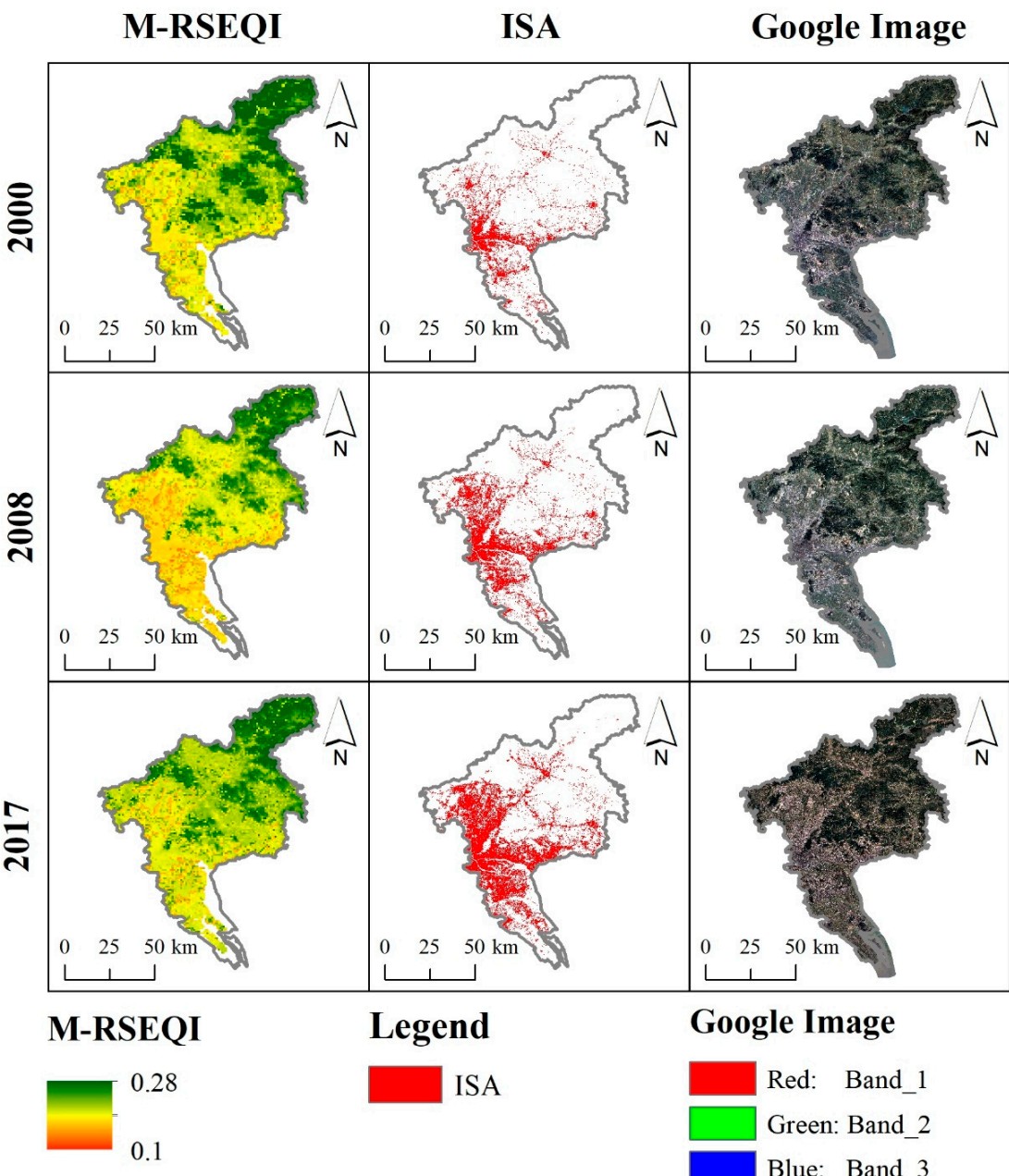

**Figure 13.** M-RSEQI, ISA, and Google historical images of Guangzhou in 2000, 2008 and 2017.

There was a notable decrease in EEQ in Beijing from 2000 to 2008, during which a considerable number of heavy industrial enterprises still operated, leading to persistent air pollution. However, despite a significant increase in impervious surface, there was no substantial decrease in EEQ in Beijing from 2008 to 2017. In 2008, Beijing implemented the relocation of heavy industrial enterprises, primarily led by Shougang, to Hebei and other areas in preparation for the Olympics [55]. As a result, air pollution in Beijing was mitigated during this period. Conversely, the EEQ of Shanghai continued to deteriorate from 2000 to 2017. Table 7 indicates that the growth rate of impervious surface in Shanghai during this period was 132.55 $km^2/a$, ranking second only to Beijing (149.10 $km^2/a$). This suggests that the rapid urbanization in Shanghai negatively impacted the EEQ. Over the period from 2000 to 2017, the EEQ of Guangzhou City initially exhibited a declining trend, followed by improvement (from 0.227 in 2000 to 0.208 in 2008), signifying a substantial decrease. On the other hand, Nanjing witnessed significant improvement in EEQ during the same period, largely attributed to advancements in water pollution control [56]. In recent years, all

districts and counties in Nanjing have actively engaged in water pollution prevention and mobilization efforts, effectively striving to establish a thriving green ecological riverfront economic zone.

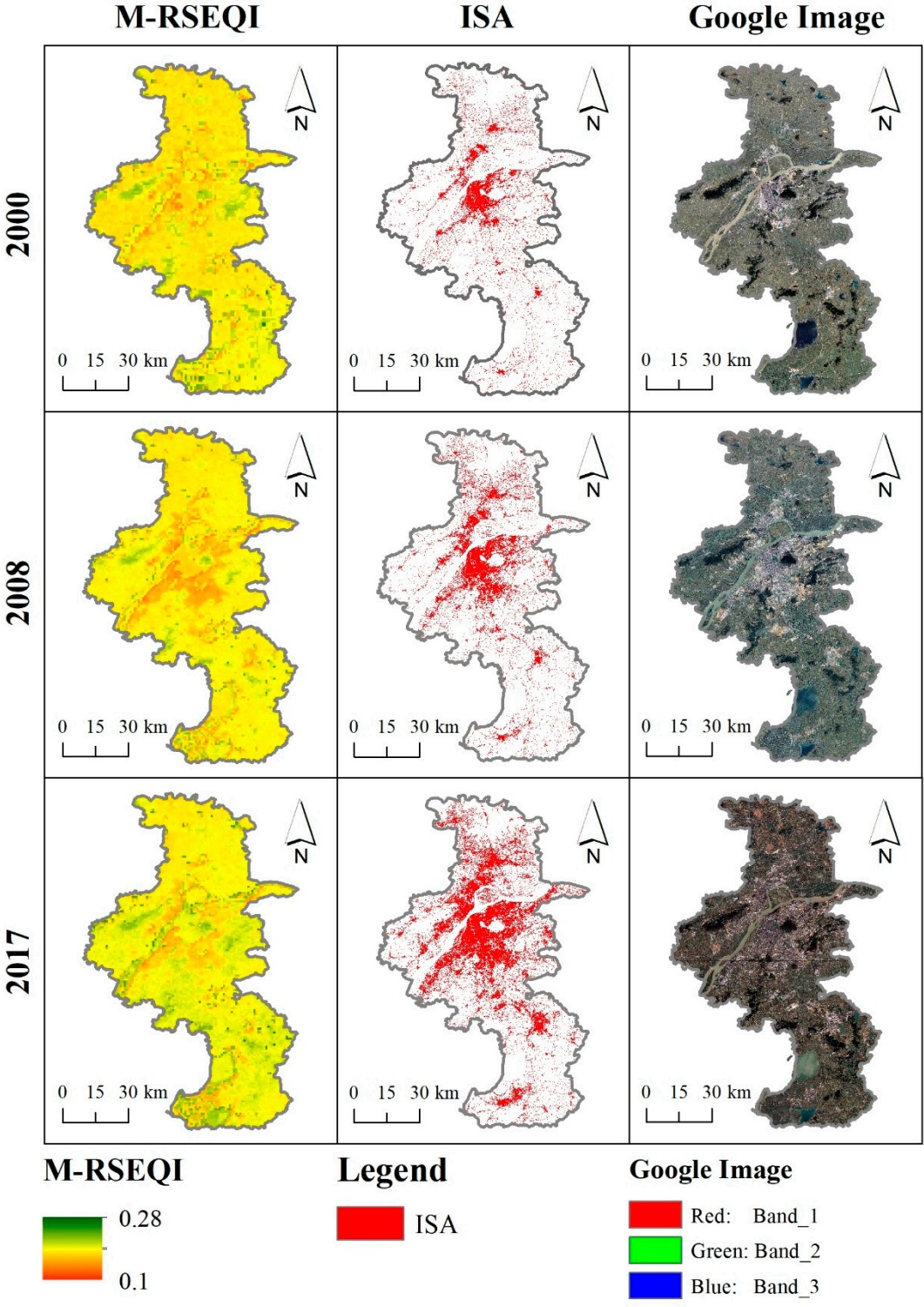

**Figure 14.** M-RSEQI, ISA and Google historical images of Nanjing in 2000, 2008, and 2017.

**Table 7.** Statistics of M-RSEQI, ISA and Speed_ISA in Beijing, Shanghai, Guangzhou, and Nanjing in 2000, 2008, and 2017.

| City | Year | M-RSEQI | ISA (km$^2$) | Speed_ISA (km$^2$/a) |
|---|---|---|---|---|
| | 2000 | 0.183 | 1961.68 | |
| Beijing | 2008 | 0.193 | 3188.98 | 149.10 |
| | 2017 | 0.193 | 4645.54 | |
| | 2000 | 0.193 | 1324.74 | |
| Shanghai | 2008 | 0.181 | 2241.25 | 132.55 |
| | 2017 | 0.187 | 3710.68 | |
| | 2000 | 0.227 | 841.00 | |
| Guangzhou | 2008 | 0.208 | 1330.20 | 61.15 |
| | 2017 | 0.223 | 1941.66 | |
| | 2000 | 0.184 | 750.25 | |
| Nanjing | 2008 | 0.183 | 1094.85 | 61.09 |
| | 2017 | 0.190 | 1849.78 | |

## 4. Discussion

In contrast to prior investigations, this study conducted long-term EEQ (Ecological and Environmental Quality) monitoring in the Chinese region using remote sensing data. This approach effectively addresses the limitations inherent in assessments based solely on statistical data [57]. For instance, Chen et al. [58] and Lv et al. [59] employed panel data to evaluate the ecological and environmental quality of China. However, the subjectivity of the statistics and the utilization of data from multiple sources led to controversial findings. By adopting this approach, the inherent constraints associated with assessments relying solely on statistical data are effectively addressed. Furthermore, by utilizing scenario data generated through Earth system models and employing the grid-scale approach proposed in this study, it becomes feasible to forecast future ecosystem quality in China and investigate its evolving attributes. We integrated the Shared Socioeconomic Pathway (SSP) [60] and Representative Concentration Pathway (RCP) [61] frameworks to establish three widely employed scenarios. These scenarios were employed to examine the spatial and temporal variations of Ecosystem Environmental Quality (EEQ) in China throughout the 21st century. The analysis was based on comprehensive remote sensing data from multiple sources and data from the Coupled Model Intercomparison Project 6 (CMIP6) model [62].

Figure 15 illustrates the consistent spatial pattern between the simulated future EEQ for China and historical EEQ, indicating the robust generalizability of the EEQ simulations conducted in this study. In Figure 15e,f, it can be observed that the average future EEQ for China exhibits a slight upward trend across the three scenarios, but the change is not pronounced. It is noteworthy that under the SSP5-RCP85 scenario, China's EEQ demonstrates a stable increasing trend (Figure 15f), implying that the country's EEQ would remain relatively unaffected by traditional fossil fuel-based pathways. This finding suggests that future human activities would have minimal impact on national EEQ.

However, it is important to note certain considerations within this study. For instance, due to limited ground validation data, it is not possible to provide an accuracy assessment for the entire study area. Furthermore, discrepancies among multiple sources of spatial data introduce some level of uncertainty into the EEQ assessment results, an aspect that was not extensively explored in this study. Nevertheless, this study offers a fresh perspective on global grid-based EEQ monitoring. Moving forward, our research endeavors will focus on investigating EEQ dynamics in rapidly urbanizing countries or regions worldwide, with particular attention to nations such as India and the USA.

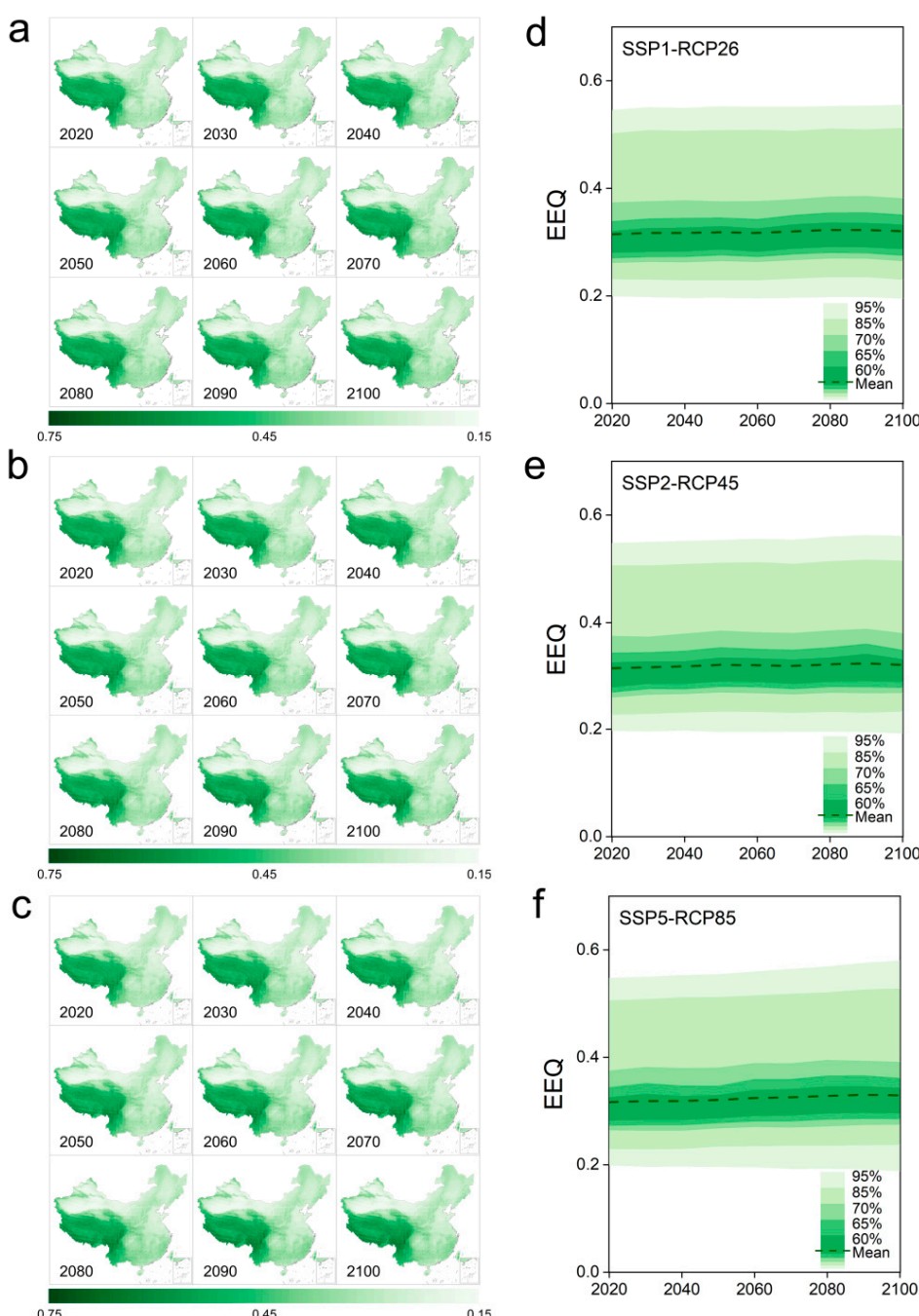

**Figure 15.** Characteristics of future (2020–2100) spatial (**a**–**c**) and temporal changes (**d**–**f**) in EEQ in China.

## 5. Conclusion

In conclusion, this research paper aimed to address the lack of comprehensive studies on the evaluation of eco-environmental quality (EEQ) in China, particularly in terms of monitoring its spatial and temporal changes at a national scale. The study highlighted the limitations of existing evaluation methods based on statistical data and proposed a novel approach using remote sensing data to construct an EEQ evaluation system, called the modified remote sensing eco-environmental quality index (M-RSEQI). The main findings of this research provide valuable insights into the EEQ trends in China from 2000 to 2017. The results showed that the EEQ of most regions in China has been improving over the past 18 years, with 60.3% of the areas experiencing an increase in M-RSEQI. However, the overall EEQ of China remained low during this period. The study also analyzed the spatial

and temporal changes in the Yellow River Basin, Yangtze River Basin, and four major cities (Beijing, Shanghai, Guangzhou, and Nanjing). The findings revealed variations in the EEQ trends among these regions, with improvements observed in the Yangtze and Yellow River basins. Additionally, the study highlighted the influence of industrial structure and urbanization on the EEQ of Beijing, with a significant reduction followed by a stable trend. Guangzhou exhibited the best EEQ among the four cities, while Shanghai experienced continuous deterioration. Notably, Nanjing's EEQ showed steady improvement due to extensive efforts in water pollution prevention and control. Overall, this research contributes to the understanding of China's ecological environment and provides objective, scientific, and reliable data support for ecological protection and pollution prevention policies, thereby helping to mitigate potential ecological risks associated with future urbanization in China.

**Author Contributions:** Conceptualization, S.W.; Methodology, S.W.; Software, S.W., D.X. and C.Z.; Formal analysis, D.X.; Resources, S.W.; Data curation, S.W.; Writing—original draft, S.W.; Writing—review & editing, L.C., D.X. and C.Z.; Supervision, L.C.; Project administration, D.X.; Funding acquisition, L.C. All authors have read and agreed to the published version of the manuscript.

**Funding:** This work was partly supported by the Key Laboratory of Environmental Change and Natural Disaster of Ministry of Education, Beijing Normal University (Grant No. 2022-KF-13).

**Institutional Review Board Statement:** Not applicable.

**Informed Consent Statement:** Not applicable.

**Data Availability Statement:** Data sharing is not applicable to this article.

**Conflicts of Interest:** The authors declare no conflict of interest.

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
