# Peer review of "Historical Eco-Environmental Quality Mapping in China with Multi-Source Data Fusion"

_applsci, doi:10.3390/app13148051_

Round 1

Reviewer 1 Report

Thank you for submitting your manuscript to the Applied Sciences journal. Generally, the topic fits into the scope of the journal, however the structure
doesnt respect Scientific Best Practice as the discussion section is missing.
The discussion section is a fundamental part of scientific work and must be
added. The abbreviation EEQ must be explained in the abstract. Moreover, the
manuscript contains only information on China which is not enough base for
an international scientific publication.
In the literature review, it is important that the scientific novelty of the work is established through a critical analysis of related literature. With this, followng questions must be clarified: How does the present work contribute towards the gaps identified? How does it improve upon previous work? It is recommended that a short discussion of the novel contribution of each reference cited shall be provided to give readers a better understanding of their relevance. Thus, the main questions of the reviewer are: What is the scientific motivation for the study? What is your scientific hypothesis that you wish to answer with the inbestigation? Putting the scientific motivation will also help you to identify the novelties that characterises a scientific publication. Moreover, the introduction contains only information on China.
The literature review must be substantially improved and expanded to the
whole world.

For the methodology section, I recommend to include a flow chart illustrating the steps of the methodology in the beginning of the methodology section. After this, all applied scientific methods need to be explained in detail.

The discussion section must be added and contain comprehensive content.
It is not enough to cut parts from the results section and create a discussion
section from this.
In the conclusions, in addition to summarising the actions taken and results, please strengthen the explanation of their significance. It is recommended to use quantitative reasoning comparing with appropriate benchmarks, especially those stemming from previous work.
Particularly information from outside china must be considered here.

The manuscript is totally China-focused. The content must be put in an international context.

Author Response

Dear editor

Re: Manuscript reference No.applsci-2413140

We sincerely thank you and the reviewers for providing us with such a valuable revision opportunity. Thus, we can further improve and present our studies. The comments from you and the reviewers were highly insightful and enabled us to greatly improve the quality of our manuscript. We seriously responded to every comment from reviewers and editors. Please note that these resulting revisions did not change the paper findings.

The submitted materials include reply letter to editor and reviewers, a “track changes” manuscripts and a “clean version” manuscript, which we would like to resubmit as a research paper for publication in Applied Science. In the “track changes” manuscripts document entitled “applsci-2413140 - the revised manuscript track version.docx”, each change we have made were tracked through the manuscript. In the reply letter, we then itemized response to editors, in which the blue font indicates the response to each comment.

We hope that the revisions in the revised manuscript and the responses to the comments will suffice to allow our manuscript to be suitable for publication in Applied Science.

Yours sincerely,

Dong Xu

Faculty of Geographical Science

Beijing Normal University

Reviewer 2 Report

The article is titled “Historical eco-environmental quality mapping in China with multi-source data Fusion”. The authors used a combination of remote sensing data from multiple sources and constructed an EEQ rating system (Modified Remote Session Eco-Environmental Quality Index, M-RSEQI). To this end, the authors used the method of entropy with objectivity and we use a system to monitor spatial and temporal changes in EEQ. The article is interesting but needs improvement. My comments are as follows:

- indicate the main goal of the work and specific goals, emphasize the novelty of this paper

- present the research methodology in detail, preferably in the form of a diagram. The authors state that they used various remote sensing data, but they do not refer to the sources of these data and the degree of their detail. The proposed methodology has not been sufficiently described, which makes it impossible to verify the correctness of the obtained results.

- the research area has not been described by the authors. There are no general parameters, e.g. rainfall, land height, land use for the entire area of China. The authors use the names of cities and provinces, but these objects are not marked on the map. For non-Chinese readers, the reception and analysis of the text is difficult. Each place mentioned in the text should be presented on the map.

- conclusions do not result from the content of the paper.

Technical Notes:

- the article should be adapted to the requirements of the journal, e.g. citing literature, entering formulas, table styles, etc.

- some figure captions are incorrectly titled, e.g. fig 1, fig 2, table 5

- in my opinion, the article should be extensively verified for linguistic correctness

​

Author Response

(The authors gave the same response as above.)

Round 2

Reviewer 1 Report

Thank you for providing the revised version of the manuscript. There are still issues with the manuscript, namely:

the novelty is still unclear

what exactly M-RSEQI expresses remains fully unclear

the evaluation levels are not explained (what means bad, poor, average, good, excellent?) what exactly is characterised with this assessment

the content and message of the figures 3 and 6 is completely unclear and must be explained

all illutrations in figure 4 look more or less equal

why suddenly in figure 7 the legend has a different colour than in the other figures?

In figure 11 is introduced a new abbreviation (ISA) that is not explained at all

the discussion section was added but is is inconsistent, to short and without a reasonable content

the conclusions remain vague, example: The main findings 488 of this research provide valuable insights into the EEQ trends in China from 2000 to 2017. ... Question: what exactly was improved?

There are still substantial issues with the manuscript

Author Response

Thank you for evaluating the strengths and weaknesses of this study and providing valuable comments that have been instrumental in enhancing the quality of the manuscript. We have thoroughly reviewed the concerns you raised and addressed each of your questions individually. We sincerely appreciate your time and effort in participating in the review of this manuscript. Please see attached for detailed responses.

Reviewer 2 Report

The authors have significantly improved the article in line with the reviewer's suggestions. I don't think the discussion was properly presented. The authors did not compare the methods or results obtained in their study with other studies. Other authors should be cited in the discussion. I suggest replacing chapter “4. Discussion and conclusion” with “4. Conlusions”. The discussion part should be inserted as subsection 3.5.

Author Response

(The authors gave the same response as above.)
